# Evaluation of Drying Characteristics and Quality Attributes for Microwave Vacuum Drying of Pork Skin Crisps

**DOI:** 10.3390/foods13244020

**Published:** 2024-12-12

**Authors:** Yuangang Li, Jingming Zhang, Junsheng Wang, Junpeng Ren, Chuanai Cao, Qian Liu, Xinning Huang

**Affiliations:** 1College of Food Science, Northeast Agricultural University, Harbin 150030, China; 13202359469@163.com (Y.L.); zhangjingmingx@163.com (J.Z.); wangjunsheng0624@163.com (J.W.); 18366573122@163.com (J.R.); caochuanai@neau.edu.cn (C.C.); 2College of Engineering, Northeast Agricultural University, Harbin 150030, China

**Keywords:** pork skin crisps, microwave vacuum drying, drying characteristics, by product utilization, quality attributes

## Abstract

As an important by-product of pork, pork skin can be processed into meat-based leisure food products to improve its utilization. In this study, microwave vacuum drying (MVD) technology was used to investigate the effects of microwave powers (600, 700, and 800 W) and processing duration on the drying characteristics and quality attributes of pork skin crisps (PSC). Five classical drying models were used to non-linearly fit the experimental data, and the Midilli et al. model was suitable for characterizing the MVD process of PSC. Before reaching a constant rate of drying, increasing microwave power and time can improve the brittleness and expansion ratio of PSC. In the constant rate drying stage, most of the free water in PSC was removed, showing the best brittleness and a stable expansion ratio. High power and long processing time can lead to serious lipid oxidation and change the flavor of PSC. Overall, the desired quality of PSC is recommended as 700 W for 6 min. This study can provide a reference for MVD application of meat-based by-product leisure foods.

## 1. Introduction

Pork is currently the most popular raw meat, and the sales volume of various pork products is also far ahead in the world and has become the second largest meat category [1]. However, there is still a big gap in the utilization rate of pork by-products compared with beef and mutton. However, the food plasticity of pig by-products is stronger than that of beef and mutton [2,3]. Pork skin is an indispensable part of pork by-product products and comes from a wide range of sources, and its price is low. Pork skin contains a large amount of collagen, which can improve the physiological function of the human body, and has a good water storage capacity to keep the skin moist and prolong aging. Leisure food has gained increasing attention for consumers, such as beef jerky and pork jerky [4]. Therefore, making pig skin into puffed leisure foods can not only meet the needs of consumers for easy-to-eat food but also provide high quality and high protein.

Currently, the processing methods of most puffed products are still traditional, such as frying puffing, hot air drying, extrusion puffing, etc. [5,6,7]. Deep-fried puffing is a method of rapidly puffing products through high-temperature frying, but deep-fried foods have high fat content and calories, and long-term consumption is harmful to human health. Excessive processing can lead to the loss of nutrients, greatly increasing processing time and energy costs [5]. Hot air drying can achieve the effect of puffing by removing moisture, but its puffing time is long and the operation is inconvenient. Long-term hot air drying will make the texture worse and the flavor defective of final leisure foods [8]. Extrusion puffing is achieved by high pressure, high temperature, and shear force to obtain high product quality. Although the puffing time is short, the processing costs are high and the heat loss is large [9]. To address the mentioned issues for the above processing methods, MVD technology may have great potential for processing PSC.

Microwave drying is to generate temperature differences through the electric-field-driven action of microwaves on polar molecules such as water, so that the water in the sample evaporates instantaneously and the polar molecules rub at a high speed to generate heat. Rapid evaporation causes water to damage muscle tissue, resulting in pores, thus expanding the material [10]. Compared with microwave drying, MVD has a negative pressure environment, which reduces the boiling point and ambient temperature of water to improve drying efficiency and reduce fat oxidation and the loss of phenolic substances [11]. Compared with the traditional expansion process, it has the characteristics of high production efficiency, less energy consumption, and less bacterial growth, mainly including ordinary microwave drying and MVD [12]. More importantly, under the action of a vacuum, the water vapor increases rapidly, which can rapidly modify the internal structure of pig skin to expand [13]. Liang et al. and Wang et al. conducted research on products such as pig and fish skin and found that microwave drying is mainly divided into two stages: the falling-rate stage and the constant-rate stage [14,15]. In the falling-rate stage, free water is rapidly removed from inside the material, while the material is mainly bound water in the constant-rate stage, which increases the resistance to water removal, and the quality is mainly formed in this stage [14].

MVD combines the common characteristics of vacuum drying and microwave drying and has been studied in fruits and vegetables, grain, and meat products [16]. Gaikwad et al. studied the drying characteristics of Indian leisure foods like black papaya and proved that MVD technology takes the least time compared with other drying technologies [17]. Ishibashi et al. measured the volume shrinkage of carrots and potatoes and confirmed that the drying rate increased with the decrease in moisture content before reaching the critical one [18]. Liu et al. reported the drying of garlic slices by MVD, evaluated its quality characteristics, and compared the difference between the dried garlic and the original sample by using the electronic nose, indicating that the vacuum and microwave methods had no effect on the original flavor of garlic [19]. Xu et al. carried out an MVD experiment on Tremella fuciformis and found that the Midilli model was the best fitting result among five kinetic models [8]. Sanchez et al. examined the effect of MVD on aroma compounds and sensory quality of rosemary and demonstrated that increasing microwave power or reducing vacuum value will increase the content of aroma compounds of rosemary within the standard range so as to improve its sensory quality [20]. Zheng et al. found that in the microwave process, the expansion ratio and brittleness are the considerable factors [21]. As the microwave power increases, the expansion ratio and brittleness of the material will increase to improve the quality and flavor of the product. Therefore, compared with microwave drying, MVD improves the brittleness and expansion ratio of products to meet the needs of consumers for expanded leisure food.

Therefore, under the condition of MVD, processing pork rinds into crispy, delicious, nutritious, and healthy ready-to-eat leisure food can not only solve the problem of single product variety of pork rinds but also enrich the varieties of puffed leisure food and further promote the development of the deep-processing industry of pork rinds. Analyze the drying characteristics and quality properties of PSC under different processing parameters. Store the completed PSC in a sealed plastic bag for further analysis.

## 2. Materials and Methods

### 2.1. Pre-Treatment of PSC

In this study, the skin of the pig spine served as the raw material. The samples of pig skin were frozen at 18 ± 1 °C after being obtained from Beida Raw Meat Products Co., Ltd. in Harbin, China. Prior to MVD, the pig sample was defrosted at 4 ± 1 °C for 5 h, and the skin of the pig was then cooked at 85 °C for 10 min in hot water. Put the precooked pig skin on a clean kitchen board and remove the grease with a kitchen knife. The pig skin was cleaned with warm water and trimmed with a circular die with a diameter of 5 cm to obtain pig skin with a diameter of 5 cm and a thickness of 2 cm. The round pig skin was boiled in 85 °C water for 30 min. The specific formula for steaming pig skin is as follows: add 0.2% salt, 0.05% monosodium glutamate, 0.2% sugar, 0.3% pepper, 1% cumin powder, 0.05% ginger powder, 0.05% onion powder, 0.05% garlic powder, and 0.05% chicken powder. The cooked pig skin was laid neatly on the baking tray and put into the oven to bake at 85 °C for 3 h until the surface is dried.

### 2.2. Microwave Vacuum Drying Experiments

Arrange the pigskin neatly in the tray, as shown in Figure 1. Conduct MVD experiments in a stationary state. The MVD experiments were carried out in a microwave vacuum dryer (ORW1.0S-5Z, Orient Microwave, Nanjing, China) with a microwave frequency of 2450 ± 10 MHz and vacuum degree of −0.093 kPa. The microwave powers (600, 700, and 800 W) and drying times were used as experimental variables to examine the drying characteristics and quality attributes of PSC. According to the preliminary experiments, the total drying time was determined as 10, 8, and 6 min for 600, 700, and 800 W, respectively, and the corresponding time interval was 2, 1, and 1 min.

### 2.3. Drying Characteristics

#### 2.3.1. Water Activity

An essential sign for preventing food corruption is water activity. A water activity meter (AquaLab 4TE, Instrument Group, Pullman, Washington, DC, USA) was used to determine the water activity of PSC samples under different drying conditions.

#### 2.3.2. Moisture Content

Dry the completed PSC in a 110 °C oven for at least 17 h, take samples every hour, weigh it until it remains stable, and obtain the moisture content (*MC*) of the PSC as determined by Equation (1) [14].
(1)$$MC=\frac{m_{t}-m_{d}}{m_{d}}$$
where *m_t_* is the moisture content of PSC before drying, and *m_d_* is the moisture content of PSC after drying.

#### 2.3.3. Moisture Ratio

The moisture ratio (*MR*) of the PSC during MVD was determined by Equation (2) [14].
(2)$$MR=\frac{M_{t}-M_{e}}{M_{0}-M_{e}}\approx \frac{M_{t}}{M_{0}}$$
where *M_t_* is the moisture content at the drying time; *M*_0_ is the initial moisture content; and *M_e_* is the equilibrium moisture content, which is often overlooked due to its small value compared to *M_t_* and *M*_0_.

#### 2.3.4. Drying Rate

The drying rate (*DR*) of PSC during MVD was calculated by using Equation (3) [22].
(3)$$DR=\frac{M_{t_{1}}-M_{t_{2}}}{t_{1}-t_{2}}$$
where *t*_1_ and *t*_2_ are drying times (min), and *M*_*t*1_ and *M*_*t*2_ are the moisture content at *t*_1_ and *t*_2_, respectively.

#### 2.3.5. Effective Moisture Diffusion Coefficient

In order to further analyze the drying characteristics of PSC, the effective moisture diffusion coefficient (*D_eff_*) that indicates the water transfer rate during MVD is used to estimate the removal of water per unit time. The *D_eff_* of PSC was calculated as Equation (4) [23].
(4)$$MR=\frac{6}{\pi^{2}}\sum_{n=1}^{\infty }\frac{1}{n^{2}}\exp\left(\frac{-n^{2}\pi^{2}D_{eff}t}{b^{2}}\right)$$
where *MR* is the moisture ratio; *D_eff_* is the effective moisture diffusion coefficient; *b* is the equivalent radius of PSC; and *t* is the drying time.

Due to the linear relationship between the logarithmic ln*MR* of the *MR* and *t*, using lnMR as the ordinate and drying time *t* as the abscissa, a linear regression fitting can be performed to obtain the slope *k* of the straight line and then calculate the *D_eff_* according to Equation (5).
(5)$$D_{eff}=\frac{-b^{2}}{\pi^{2}}k$$

#### 2.3.6. Kinetics Modeling

In order to quantitatively describe the drying characteristics of PSC, as shown in Table 1, the moisture ratios under different microwave intensities were fitted and analyzed using five traditional thin-layer drying models [24].
(6)R2=Σⅈ=1NMRexp,i−MRpre,i2Σⅈ=1NMRexp,i−MRpre,i¯2
(7)χ2=1−Σi=1NMRexp,i−MRpre,i2N−n
(8)RMSE=Σⅈ=1NMRexp,i−MRpre,i2N
where *MR*_*pre*,*i*_ and *MR*_*exp*,*i*_ represent the predicted and measured values of *MR* at the data points, respectively; *N* represents the number of test data points; *n* represents the number of test points in the model under test; and MRexp,i¯ is the average *MR* projected value at the data point.

### 2.4. Quality Characteristics

#### 2.4.1. Color

Pig skin’s color was ascertained using a colorimeter (Nippon Denshoku, kogyo Co., Ltd., Tokyo, Japan). Cao et al. stated that the approach described is somewhat altered [25]. The color index is represented by *L**, *b**, and *a**, respectively. Color *L** represents brightness and darkness, 0 (black) and 100 (white). Color a* represents red (+) and green (−), while color *b** represents yellow (+) and blue (−). Measure the color of PSC at different times under 600 W, 700 W, and 800 power. The color value of each sample is determined 6 times.

#### 2.4.2. Determination of Brittleness

The texture characteristic parameters of PSC were determined by a texture analyzer (XT2i/50, Stable Micro Systems Co., Godalming, UK). The P/5 cylindrical probe with a diameter of 5 mm was used to test the sample. The test speed was 2, 0.1, and 2 mm/s before during and after testing, respectively. The Exponent software (Ver. 6.1.16.0) equipped with the instrument was used to calculate the brittleness of the product.

#### 2.4.3. Determination of the Expansion Ratio

The volume replacement method was used to determine the volume before and after expansion. Briefly, taking a certain amount of about 20 mL of quartz sand into a measuring cylinder (accuracy of 2 mL), putting in the tested PSC, and then adding quartz sand, which is about 20 mL higher than the measured object, and putting it on an oscillator to vibrate until the height does not drop. Read out the total volume *V*_1_, take out the fruit slice, and also measure the volume *V*_2_ of quartz sand; then, the volume *V*_3_ before and after expansion can be calculated by Equation (9) [26].
(9)$$V_{3}=\frac{V_{1}-V_{2}}{V_{2}}$$

#### 2.4.4. Electronic Nose

The electronic nose test was conducted using the method described by [27], and the PEN 3.5 electronic nose (Win Muster Airsense Analytics, Schwerin, Germany) was used. The PEN 3.5 system consists of 10 metal oxide gas sensors, named W1C, W5S, W3C, W6S, W5C, W1S, W1W, W2S, W2W, and W3S, respectively. These sensors have differential sensitivity for each characteristic volatile compound. The performance descriptions of each sensor were shown in Table 2. Finally, the principal component analysis (PCA) method was used to analyze the electronic nose data.

#### 2.4.5. Measurement of Lipid Oxidation

Malondialdehyde (MDA) content was used to evaluate the lipid oxidation, and it was determined using a slightly modified version of the method [28]. Moreover, 1.0% TBA solution, 3.0 mL, 2.5% TCA-HCl solution, and 2.0 g of PSC should be combined. Then, vortex the mixture using a vortex mixer (3030A, Scientific Industries, INC., Bohemia, Czech Republic). After heating for 30 min, the swirling mixture was cooled to room temperature. Furthermore, 4.0 mL of the obtained supernatant was combined with 4.0 mL of chloroform, followed by centrifugation at 3000 rpm for 10 min. Finally, a UV visible spectrophotometer (T6 New Century, Beijing Puxi, Beijing, China) was used to measure the absorbance at 532 nm. The TBA reactive substance (*TBARS*) value was calculated by Equation (10).
(10)$$TBARS\;(\mathrm{mg}/\mathrm{kg})=\frac{OD_{532}}{m}\times 9.48$$
where *OD*_532_ represents the absorbance at 532 nm; *m* represents the weight of the sample (g); and the constant 9.48 is a fixed value used in the calculation.

#### 2.4.6. Sensory Evaluation

Sensory evaluation of PSC under different drying conditions was conducted using Chen’s method [29]. The number of PSC participants is 20, including 10 females and 10 males. Before conducting sensory evaluation tests, food professionals provided professional training to 20 students. Evaluate PSC as follows: color (1 = black, 7 = golden yellow), flavor (1 = obvious oxidized flavor, 7 = no odor), appearance (1 = no puffing at all, 7 = completely puffing), brittleness (1 = hard, 7 = crispy) (Table 3). Place the completed PSC in a randomly numbered sample tray and immediately send it to 20 students for sensory evaluation analysis.

### 2.5. Statistical Analysis

The PSC samples were prepared into three independent batches (repeated), and three replicates of each batch were run. The results were expressed as mean ± standard error (SE). SPSS statistical software (Ver. 22.0, IBM SPSS Co., Chicago, IL, USA) was used for data analysis. Two-way analysis of variance (ANOVA) and Duncan multivariate range test were used (*p* < 0.05). Data were plotted using Origin Software (Ver. 2022, Origin lab Corporation, Northampton, MA, USA).

## 3. Results

### 3.1. Drying Characteristics of PSC Under MVD

#### 3.1.1. Changes in Water Activity of PSC During MVD

As shown in Table 4, the water activity of PSC during MVD first showed a rapid decline trend, followed by a slow decline trend when the water activity reached about 0.25. This type of feature has also appeared in the microwave process of pork crisps, which is called the rapid drying stage and the hindered drying stage [30]. Among microwave intensities, high-power microwave intensity can quickly make the PSC enter the hindered drying stage. This indicated that the water in the PSC evaporated rapidly during the rapid drying stage, and the water removal process was very fast. This was due to the high-speed heating of PSC with high moisture content, which caused PSC to absorb and convert higher microwave energy, resulting in significant dehydration of PSC [31]. In the hindered drying stage, although the water activity decreased from 0.25 to 0.23, the drying rate became significantly slower. This was because the moisture content in the sample was already in a low state. Although the microwave power remained unchanged, the drying rate of the sample decreased rapidly, and it was difficult to remove the remaining water, resulting in a declining rate of water activity. It was at this point that product quality began to take shape, and the change in sample quality should be noted at this stage.

#### 3.1.2. Changes in Drying Rate of PSC During MVD

Figure 2a shows the change in the drying rate of PSC with drying time from 600 to 800 W, and the drying rate also increased slowly with the increase in microwave intensity. It was obvious that the drying rate of PSC changes the most with drying time at 800 W and fluctuates locally at 600 and 700 W, which may be due to the pressure difference between the periphery and center of the PSC during the drying process, resulting in unstable and fluctuating moisture transfer. It can also be seen that the drying rate reaches its maximum at the beginning of drying. With the extension of drying time, the drying rate decreases slowly. Such phenomena have also occurred in germinated brown rice and apple slices [31,32]. This may be due to the movement of water molecules under the influence of variable electromagnetic fields at the beginning of drying. Due to friction with adjacent molecules, heat is generated, which can remove moisture from PSC [33].

#### 3.1.3. Changes in Moisture Ratio of PSC During MVD

From Figure 2d, it can be seen that under the action of 600 W, 700 W, and 800 W power, the moisture content of PSC decreased from the initial 6.1657 to 0.3147, 0.2847, and 0.2764, respectively. Figure 2b shows the change in moisture ratio with drying time under 600 to 800 W. With the extension of drying time, the moisture ratio gradually decreased, reaching equilibrium at 600 W for 8 min, 700 W for 6 min, and 800 W for 4 min, respectively. The higher the microwave power, the faster the moisture ratio decreased. This may be because high microwave power enables PSC to absorb more energy, thereby creating a pressure difference between the inside and outside of the PSC to reduce surface moisture and increase the drying rate [34,35]. Therefore, increasing microwave intensity can quickly remove water and shorten drying time.

#### 3.1.4. Changes in Effective Moisture Diffusion Coefficient of PSC During MVD

Table 5 shows that the estimated value of *D_eff_* was between 2.3695 × 10^−5^ and 3.3395 × 10^−5^ when the microwave intensity was between 600 and 800 W. In addition, the *D_eff_* value increases with the increase in microwave intensity. This was because in the same drying time, the material increases the volume heat generation under higher microwave intensity, which led to the enhancement of the activity of water molecules. This also confirms the above viewpoint [36]. Therefore, increasing the microwave intensity can accelerate the diffusion of water molecules in the PSC because the drying driving potential of water increased with the increase in temperature, which promoted the removal of water and caused the correspondingly high *D_eff_* value. This was also reflected in eggplant slices [37].

#### 3.1.5. Analysis of Drying Kinetics of PSC During MVD

To further explain the kinetics of PSC in the MVD process, the moisture ratio of the PSC was modeled using five classical models listed in Table 1. According to previous research, the drying model should have higher R^2^ and $\chi^{2}$ as well as lower RMSE [38]. Table 6 shows that the ranges for the R^2^, $\chi^{2}$, and RMSE were, respectively, 0.9154 to 0.9984, 0.0003 to 0.0328, and 0.0173 to 0.0991. Based on the above results, the Midelli et al. model was more suitable for characterizing the MVD process of PSC.

### 3.2. Drying Quality Characteristics of PSC Under MVD

#### 3.2.1. Analysis of Color Characteristics of PSC

Color is an important visual index of PSC after MVD, and it characterizes the appearance quality. Table 4 shows that with the increase in microwave power and the extension of drying time, the *a** and *b** values of PSC increased, and the *L** value decreased. It showed that when the hindered drying stage was just reached, the PSC was cash yellow, and with the extension of time, the PSC was burnt yellow and scorched. This may be due to the rapid removal of moisture during the rapid drying stage, where the PSC gradually started to mature and slowly turns golden yellow from milky white [39]. However, during the hindered drying stage, due to the low moisture content in PSC, the fat began to oxidize and the protein began to denature, while the high microwave intensity and long drying time accelerated the fat oxidation and protein denaturation, resulting in the appearance of burnt yellow [14].

#### 3.2.2. Changes in Brittleness of PSC

Figure 3a shows the changes in brittleness of PSC under different microwave intensities. The brittleness of PSC increases first and then decreases with the extension of drying time. The brittleness of the final PSC was greater at 600 W for 7 min, 700 W for 6 min, and 800 W for 4 min, respectively. Combined with the changes in moisture content (Figure 2b), the brittleness of PSC reached its maximum at the beginning of the hindered drying stage. For different microwave intensities, the brittleness dried at 700 W for 6 min and reached the maximum value. This may be because with the increase in microwave intensity and drying time, the moisture content in PSC was low, the protein in PSC was denatured, and the secondary structure and tertiary structure were changed, leading to the hardening of PSC [40]. In conclusion, the brittleness of PSC improved under MVD.

#### 3.2.3. Changes in Expansion Ratio of PSC

Figure 3b shows the changes in the expansion ratio of PSC. The maximum expansion ratio of PSC was about 200%. With the extension of drying time, the expansion ratio of PSC slowly increased and finally tended to be stable. The higher the microwave intensity, the faster it reached the maximum expansion ratio. At the beginning of the hindered drying stage, the expansion ratio of PSC reached a stable stage. The expansion of PSC was mainly caused by the microwave puffing effect due to the large pressure difference between the inside and outside of the material under a microwave field [41].

#### 3.2.4. Analysis of TBRAS

Lipid oxidation was one of the effects on sensory properties of meat products, and TBARS was an important index to detect secondary lipid oxidation. For the lipid oxidation of by-products such as PSC, TBARS was the most intuitive indicator that can be expressed within a certain range. It can be seen from Figure 4 that under the same microwave power, TBARS increased continuously with the extension of drying time and storage time. This was because with the continuous increase in drying temperature, the quality structure of the material was broken, resulting in significant changes in the degree of fat oxidation. Compared with Figure 4a–c, with the increase in microwave power, its TBARS also increased. It may be that heme iron releases free iron, which reacts with free iron in PSC and damages lipids and proteins through oxidation [42].

#### 3.2.5. Detection of the Electronic Nose and PCA

It can be seen from Figure 5 that the response values of W1S (alkanes), W2S (alcohols), and W6S (hydrides) rank among the top three, and the top three response values at each power were the same, so the types of aroma substances of PSC were similar for different microwave powers [43]. However, with the increase in drying time, the response value basically increased gradually. In the MVD process, the increasing drying time can lead to the production of various components, the most obvious of which were alkanes, alcohols, and hydrides. It can be seen from Figure 5a,b that the hydride value of 600 W for 10 min was higher than that at 700 W for 8 min, which corresponded to TBARS. The interior temperature of PSC increased with the expansion of duration under 600 W, which caused fat oxidation.

The response values obtained through the electronic nose were used for PCA of PSC under different conditions. Figure 6 shows the PCA results at 600, 700, and 800 W, respectively. Under different microwave powers, the two principal components accounted for approximately 91.6%, 82.9%, and 93% of the total variance, with the first and second principal components accounting for 69.5% and 22.1%, 60.2% and 22.7%, 80.9% and 12.1%, respectively. Therefore, the first principal component was used for the analysis. As shown in Figure 6a, under 600 W, except for W1C, W3C, and W5C, all response values were positively correlated with the drying time of PSC. As shown in Figure 6b, under 700 W, except for W1C, W3C, W5C, and W2W, all response values were positively correlated with the drying time of PSC. As shown in Figure 6c, under 800 W, except for W1C, W3C, W5C, and W2W, all response values were positively correlated with the drying time of PSC. For the second principal component, W1S, W2S, and W1C all showed a negative correlation at different microwave powers, but W2W showed a negative and positive correlation under 700 and 800 W, respectively. Therefore, the combination of PC1 and PC2 can reflect the differences in the response values of PSC under different microwave powers and drying times [44].

#### 3.2.6. Sensory Evaluation

Table 7 shows the effects of different drying conditions on the color, flavor, appearance, and brittleness of PSC. It can be seen from the table that the color and flavor of PSC gradually decrease with the extension of drying time and the continuous increase in drying power, which is consistent with the results of color difference, TBARS, and electronic nose. This indicates that with the extension of drying time and the continuous increase in drying power, the degree of PSC oxidation gradually increases and the flavor changes. However, with the extension of drying time and the continuous increase in drying power, the score of brittleness and appearance showed a trend of first increasing and then decreasing. The highest score appeared after drying for 6 min at 700 W power, which is roughly the same as the result of brittleness.

## 4. Conclusions

This study investigated the effects of different microwave powers and drying times on the drying characteristics (water activity, moisture ratio, drying rate, effective water diffusion coefficient) and quality attributes (color, brittleness, expansion ratio, electronic nose, fat oxidation) of PSC under MVD conditions. The results showed that the Midilli et al.’s model was found to be fit the best for characterizing the MVD process of PSC among five typical thin-layer drying models. Drying time and microwave power have a significant effect on the product quality of the final PCS. In the constant-rate drying stage, the drying characteristics of PSC were stable, and the brittleness and expansion effect were the best. Excessively long time and high power will lead to the deterioration of the quality of PSC. The quality formation of PSC mainly occurred in the constant-rate drying stage, and the effect was obvious at the critical moisture point at 700 W for 6 min. The optimal processing parameter of PSC under MVD was 700 W for 6 min, and it further showed that MVD had a significant impact on the quality of pig by-product leisure food such as PSC. The product is easy to make, with low energy consumption and a short working time, making it more convenient and efficient to produce the required samples. It is suitable for processing in factories. It provides a new idea for the processing of by-products in the future. By combining microwave technology with meat products, it further expands the field of microwave puffing products and meat product processing, providing a new idea for the future processing of pig skin by-products as leisure foods.

## Figures and Tables

**Figure 1 foods-13-04020-f001:**
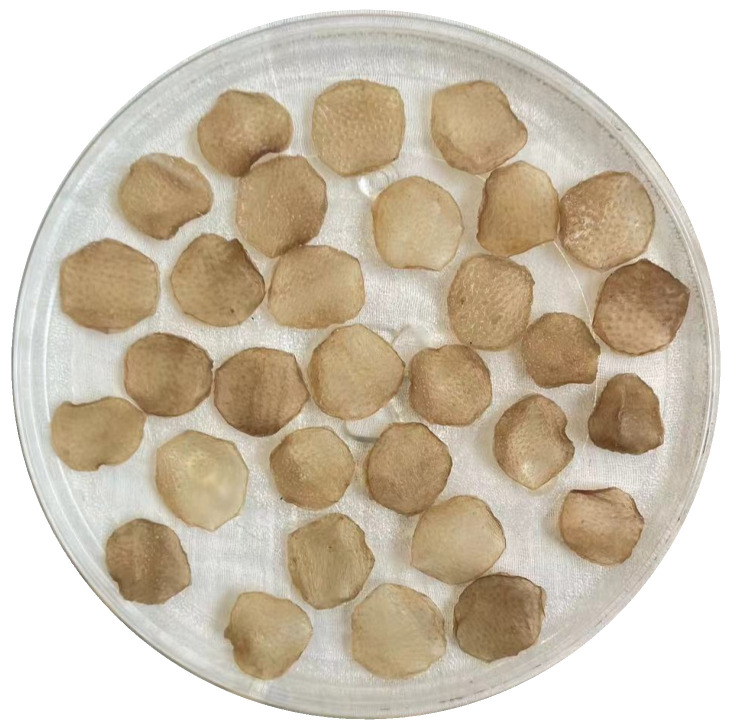
PSC tray placement before MVD.

**Figure 2 foods-13-04020-f002:**
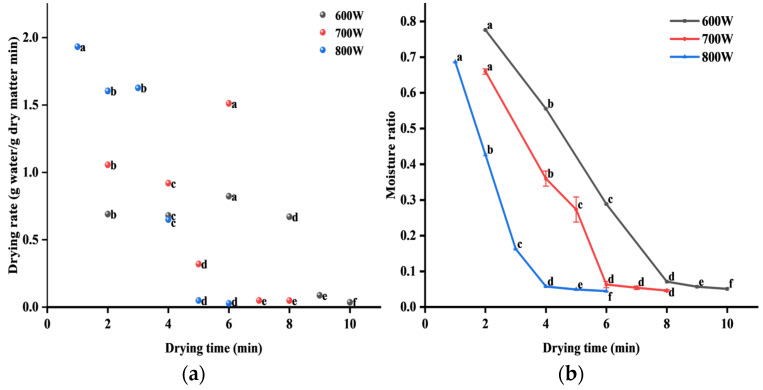
Drying curves of PSC under different microwave intensities. (**a**) The change in drying rate with drying time. (**b**) The change in moisture ratio with drying time. (**c**) The relationship between LnMR and drying time. (**d**) The change in moisture content with drying time. The letters a–f indicate the significance at *p* < 0.05 between the different treatments.

**Figure 3 foods-13-04020-f003:**
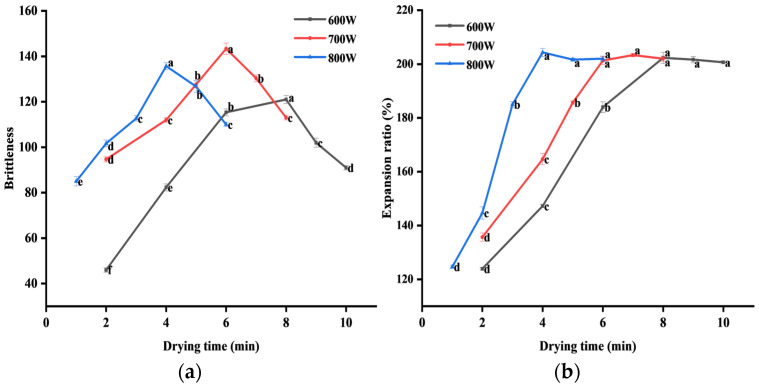
Quality changes in (**a**) brittleness and (**b**) expansion ratio of PSC with the drying time under different microwave intensities. The letters a–f indicate the significance at *p* < 0.05 between the different treatments.

**Figure 4 foods-13-04020-f004:**
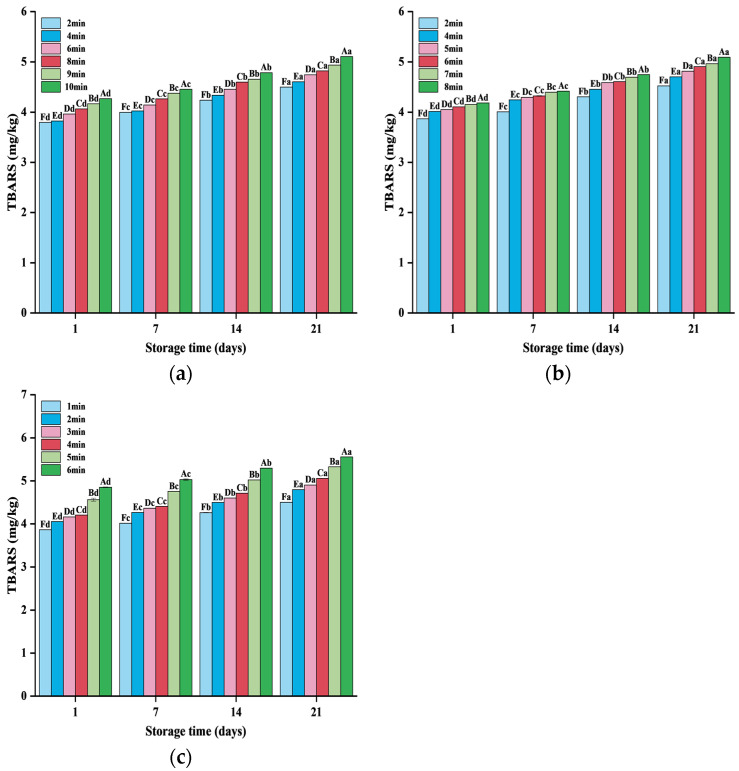
The change in TBRAS value at (**a**) 600 W, (**b**) 700 W, and (**c**) 800 W with the storage days under different drying times. The different letters indicate the significance at *p* < 0.05 between the different treatments.

**Figure 5 foods-13-04020-f005:**
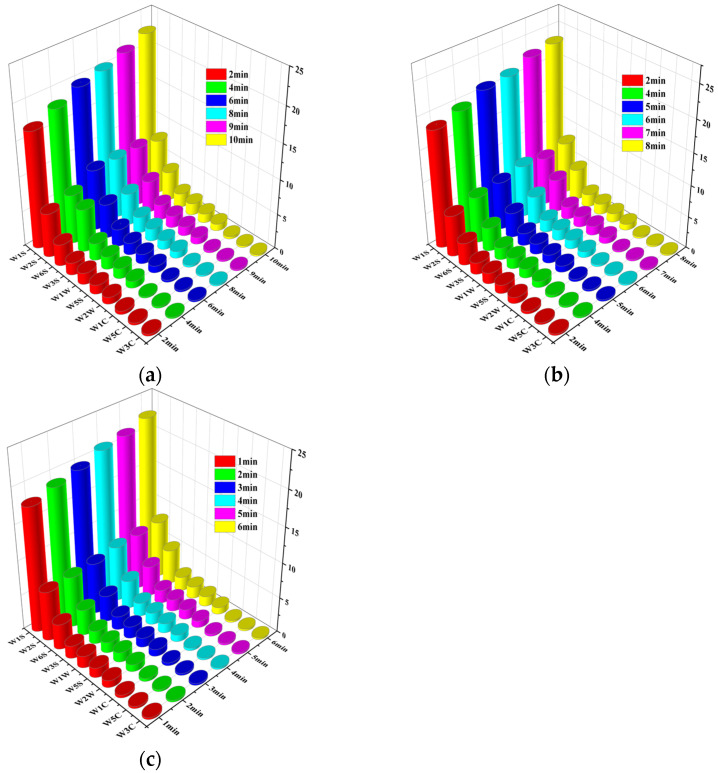
Changes in the electronic nose data at (**a**) 600 W, (**b**) 700 W, and (**c**) 800 W under different drying times.

**Figure 6 foods-13-04020-f006:**
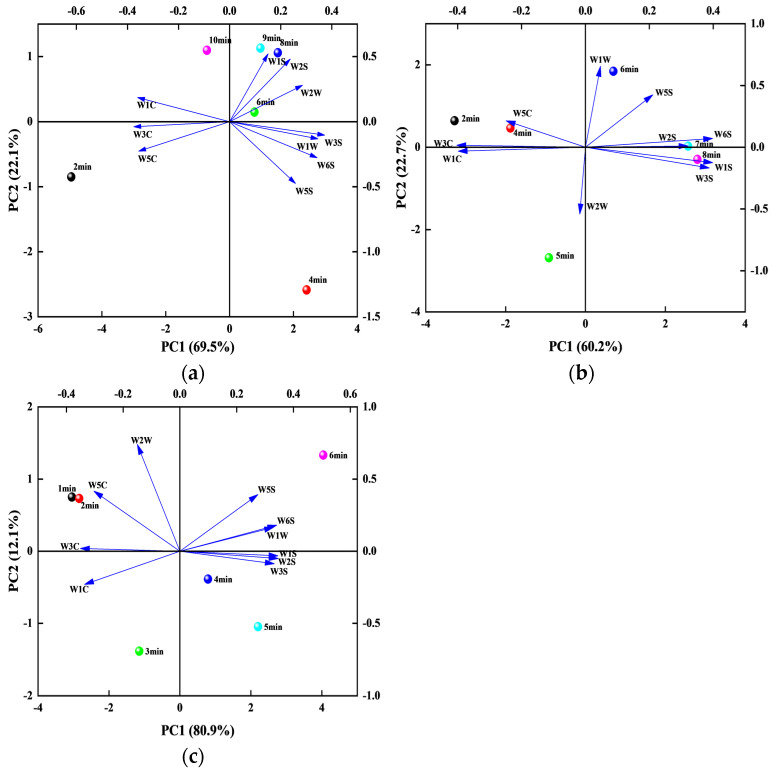
PCA of the electronic nose at different times under (**a**) 600 W, (**b**) 700 W, and (**c**) 800 W.

**Table 1 foods-13-04020-t001:** Drying models used for characterizing the MVD process of PSC.

Number	Model Name	Model Expression	Model Parameters
1	Newton	*MR* = exp(−*kt*)	*k*
2	Page	*MR* = exp(−*kt^n^*)	*k*, *n*
3	Wang and Singh	*MR* = 1 + *at* + *bt*^2^	*a*, *b*
4	Henderson and Pabis	*MR* = *a* × exp(−*kt*)	*a*, *k*
5	Midilli et al.	*MR* = *a* × exp(−*kt^n^*) + *bt*	*a*, *k*, *n*, *b*

**Table 2 foods-13-04020-t002:** Material classification of electronic nose sensor.

Number	Sensor	Substances for Sensing
S1	W1C	Aromatic compounds
S2	W5S	Nitrogen oxide
S3	W3C	Aromatic molecule
S4	W6S	Hydride
S5	W5C	Olefin, Aromatics
S6	W1S	Alkanes
S7	W1W	Sulfur compound
S8	W2S	Alcohols
S9	W2W	Organic compounds of sulfur
S10	W3S	Aliphatic group

**Table 3 foods-13-04020-t003:** Sensory scoring criteria.

Sensory Items	Scoring Criteria	Score
Color	Golden yellow color, uniform color, no burnt phenomenon	6–7
	It appears golden or brownish yellow, with a relatively uniform color and a slight burnt appearance	4–5
	Brownish yellow in color, uneven in color, almost dull, partially burnt	2–3
	Black brown in color, uneven in color, severely burnt	1
Flavor	No odor	6–7
	There is a slight smell of oxidation	4–5
	Has a strong oxidizing odor	2–3
	There is a noticeable smell of oxidation	1
Appearance	Honeycomb-like uniformity, complete and evenly distributed foaming, fully expanded	6–7
	Honeycomb-shaped uniform, with less complete and evenly distributed foaming and some puffing	4–5
	Honeycomb-like structure is relatively uniform, with incomplete foaming and uneven distribution around the edges and no puffing at the edges	2–3
	Honeycomb-like unevenness, incomplete foaming, significant size differences, and overall uneven distribution, without any puffing at all	1
Brittleness	Crispy texture and good chewiness	6–7
	Relatively crispy, with slightly an uneven texture of softness and hardness	4–5
	Partially crispy, with uneven texture of softness and hardness	2–3
	Soft or severely hardened texture, difficult to chew	1

**Table 4 foods-13-04020-t004:** Changes in water activity and color of PSC with the drying time under different microwave intensities.

Microwave Intensity (W)	Drying Time (min)	Water Activity	*L** Value	*a** Value	*b** Value
600	2	0.462 ± 0.006 ^A^	80.563 ± 0.007 ^A^	5.087 ± 0.189 ^I^	17.618 ± 0.108 ^Q^
	4	0.393 ± 0.002 ^C^	79.710 ± 0.021 ^B^	4.717 ± 0.024 ^J^	18.753 ± 0.072 ^P^
	6	0.323 ± 0.002 ^F^	79.203 ± 0.039 ^BC^	4.523 ± 0.058 ^K^	19.580 ± 0.087 ^O^
	8	0.250 ± 0.003 ^IJK^	78.863 ± 0.033 ^C^	4.403 ± 0.049 ^L^	20.097 ± 0.041 ^N^
	9	0.246 ± 0.006 ^JKL^	77.863 ± 0.033 ^D^	4.253 ± 0.026 ^M^	20.703 ± 0.113 ^M^
	10	0.239 ± 0.002 ^LM^	76.380 ± 0.015 ^E^	4.237 ± 0.034 ^M^	21.337 ± 0.112 ^AL^
700	2	0.426 ± 0.007 ^B^	75.373 ± 0.218 ^E^	5.877 ± 0.058 ^E^	21.850 ± 0.068 ^K^
	4	0.347 ± 0.003 ^E^	74.753 ± 0.041 ^F^	5.730 ± 0.057 ^F^	22.133 ± 0.062 ^J^
	5	0.296 ± 0.004 ^H^	73.887 ± 0.007 ^G^	5.503 ± 0.030 ^G^	22.657 ± 0.084 ^I^
	6	0.253 ± 0.009 ^IJ^	73.073 ± 0.050 ^H^	5.373 ± 0.021 ^H^	23.080 ± 0.064 ^H^
	7	0.243 ± 0.003 ^JKL^	72.910 ± 0.060 ^H^	5.270 ± 0.031 ^H^	23.533 ± 0.056 ^G^
	8	0.238 ± 0.002 ^LM^	71.383 ± 0.096 ^J^	5.137 ± 0.064 ^I^	24.233 ± 0.056 ^F^
800	1	0.458 ± 0.003 ^A^	72.070 ± 0.070 ^I^	6.860 ± 0.036 ^A^	24.873 ± 0.067 ^E^
	2	0.381 ± 0.001 ^D^	71.223 ± 0.088 ^J^	6.670 ± 0.064 ^B^	25.250 ± 0.055 ^D^
	3	0.309 ± 0.005 ^G^	69.807 ± 0.046 ^K^	6.490 ± 0.0292 ^C^	26.230 ± 0.121 ^C^
	4	0.258 ± 0.002 ^I^	69.127 ± 0.056 ^K^	6.423 ± 0.038 ^C^	26.317 ± 0.055 ^C^
	5	0.240 ± 0.003 ^KLM^	69.093 ± 0.060 ^K^	6.267 ± 0.074 ^D^	26.587 ± 0.029 ^B^
	6	0.231 ± 0.002 ^M^	69.323 ± 0.304 ^K^	6.193 ± 0.009 ^D^	26.980 ± 0.017 ^A^
*P* (Microwave intensity)		**	**	**	**
*P* (Drying time)		**	**	**	**
*P* (Microwave intensity × Drying time)		**	**	ns	**

Note: Data were expressed as means ± standard deviation (*n* = 3); ^A–M^ in the third column, ^A–K^ in the fourth column, ^A–M^ in the fifth column, and ^A–Q^ in the sixth column indicate significant differences (*p* < 0.05) in water activity, brightness, redness, and yellowness values among samples with different degrees of dryness. **, very important (*p* < 0.01). ns, the difference is not significant.

**Table 5 foods-13-04020-t005:** Effective moisture diffusion coefficient of PSC under different microwave intensities.

Microwave Intensity (W)	Fitted Equation	R^2^	*D_eff_* (m^2^/s)
600	LnMR = 0.6380 − 0.3738t	0.9562	2.3695 × 10^−5^
700	LnMR = 0.7209 − 0.4999t	0.9214	3.1689 × 10^−5^
800	LnMR = −0.0451 − 0.5356t	0.9390	3.3395 × 10^−5^

**Table 6 foods-13-04020-t006:** According to the drying model shown in Table 1, the modeling results of moisture ratio and drying time under different microwave intensities.

Number	Model Name	Microwave Intensity (W)	Model Parameter	R^2^	*χ* ^2^	RMSE
1	Newton	600	*k* = 0.2106	0.9734	0.2278	0.0900
		700	*k* = 0.2873	0.9794	0.1943	0.0740
		800	*k* = 0.5026	0.9807	0.0930	0.0543
2	Page	600	*k* = 0.0589 *n* = 1.7345	0.9868	0.0276	0.0289
		700	*k* = 0.1311 *n* = 1.5152	0.9805	0.0856	0.0461
		800	*k* = 0.3516 *n* = 1.4190	0.9876	0.1208	0.0245
3	Wang and Singh	600	*a* = −0.1378 *b* = 0.0038	0.9840	0.1740	0.0401
		700	*a* = −0.1972 *b* = 0.0092	0.9839	0.1778	0.0432
		800	*a* = −0.3746 *b* = 0.0360	0.9956	0.0361	0.0238
4	Henderson and Pabis	600	*a* = 1.0711 *k* = 0.2231	0.9487	0.2016	0.0848
		700	*a* = 1.0403 *k* = 0.2962	0.9567	0.1851	0.0722
		800	*a* = 1.0387 *k* = 0.5192	0.9794	0.0837	0.0519
5	Midilli et al.	600	*a* = 1.0111 *b* = 0.1197 *n* = 7.5693 *k* = −5.7674	0.9980	0.0192	0.0157
		700	*a* = 0.9870 *b* = −0.1531 *n* = 5.8489 *k* = −1.3273	0.9888	0.0552	0.0362
		800	*a* = 0.9930 *b* = 0.0046 *n* = 1.5124 *k* = 0.3391	0.9960	0.0181	0.0218

**Table 7 foods-13-04020-t007:** Sensory scoring results of PSC.

Microwave Intensity (W)	Drying Time (min)	Color	Flavor	Appearance	Brittleness
600	2	6.23 ± 0.252 ^B^	6.40± 0.100 ^A^	5.33 ± 0.058 ^CD^	3.70 ± 0.200 ^EFG^
	4	6.21 ± 0.100 ^A^	6.20 ± 0.100 ^A^	5.13 ± 0.153 ^D^	4.27 ± 0.252 ^CDEF^
	6	6.17 ± 0.153 ^B^	6.23 ± 0.208 ^A^	5.77 ± 0.321 ^B^	5.87 ± 0.252 ^AB^
	8	5.67 ± 0.115 ^C^	5.70 ± 0.265 ^CD^	6.23 ± 0.115 ^A^	6.20 ± 0.100 ^A^
	9	4.20 ± 0.100 ^E^	4.97 ± 0.208 ^E^	5.53 ± 0.208 ^BC^	4.90 ± 1.646 ^CD^
	10	2.47 ± 0.208 ^F^	3.43 ± 0.252 ^GH^	5.30 ± 0.100 ^CD^	4.20 ± 0.100 ^CDEF^
700	2	6.70 ± 0.100 ^A^	6.33 ± 0.208 ^A^	5.60 ± 0.300 ^BC^	3.33 ± 0.208 ^FG^
	4	6.40 ± 0.200 ^AB^	6.30 ± 0.100 ^A^	5.17 ± 0.058 ^D^	4.10 ± 0.100 ^DEFG^
	5	6.20 ± 0.200 ^B^	6.33 ± 0.153 ^A^	6.27 ± 0.058 ^A^	5.93 ± 0.351 ^AB^
	6	5.27 ± 0.153 ^D^	6.13 ± 0.153 ^AB^	6.50 ± 0.200 ^A^	6.67 ± 0.153 ^A^
	7	4.10 ± 0.200 ^E^	5.73 ± 0.153 ^CD^	4.73 ± 0.153 ^E^	5.20 ± 0.100 ^BC^
	8	2.13 ± 0.306 ^G^	4.63 ± 0.058 ^F^	3.50 ± 0.100 ^H^	4.17 ± 0.115 ^CDEF^
800	1	6.70 ± 0.100 ^A^	6.30 ± 0.100 ^A^	5.30 ± 0.100 ^CD^	3.33 ± 0.321 ^FG^
	2	6.40 ± 0.200 ^AB^	5.87 ± 0.153 ^BC^	5.20 ± 0.173 ^D^	4.23 ± 0.058 ^CDEF^
	3	6.20 ± 0.200 ^B^	5.43 ± 0.153 ^D^	5.73 ± 0.153 ^B^	4.67 ± 1.429 ^CDE^
	4	5.27 ± 0.153 ^D^	4.53 ± 0.289 ^F^	6.30 ± 0.100 ^A^	6.33 ± 0.153 ^A^
	5	4.10 ± 0.200 ^E^	3.73 ± 0.252 ^G^	5.30 ± 0.100 ^CD^	4.80 ± 0.100 ^CD^
	6	2.13 ± 0.306 ^G^	3.23 ± 0.153 ^H^	4.30 ± 0.265 ^G^	3.10 ± 0.300 ^G^
*P* (Microwave intensity)		**	**	**	**
*P* (Drying time)		**	**	**	**
*P* (Microwave intensity × Drying time)		**	**	**	**

Note: The different superscripted letters in each column indicate the significance at *p* < 0.05 between the different treatments. **, very important (*p* < 0.01).

## Data Availability

The original contributions presented in the study are included in the article, further inquiries can be directed to the corresponding authors.

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
