# Peer review of "Evaluation of Drying Characteristics and Quality Attributes for Microwave Vacuum Drying of Pork Skin Crisps"

_foods, 2024, doi:10.3390/foods13244020_

Round 1
Reviewer 1 Report
Comments and Suggestions for Authors
1. Line 124 drying rate , no drying ratio
2. V3 in formula 8 is not an expansion rate, it is a relative change in volume
3. Lack of storage conditions described in the methodology
4. Please state in how many repetitions MVD drying was performed. If the results are from only one experiment, without repetitions, then the results are not very reliable
5. Please state what was the water content in the peel at the beginning and after pre-treatment?
6. Drying curves should be described by models based on all experimental data from repetitions and the drying rate should be calculated as a derivative of the function from the best-fit model. The method of calculating the drying rate proposed by the authors is burdened with too large an error
7. Please justify that under microwave heating conditions, water transport is of a diffusion nature (molecular transport). In my opinion, there is a convective movement of steam forced by the pressure difference. Therefore, the law describing the phenomenon of diffusion cannot be applied
8. When using a microwave field, there is no absorption of microwaves, there is the setting of water molecules in motion under the influence of a variable electromagnetic field. As a result of friction with neighboring molecules, heat is generated
9. The authors do not determine the rate of expansion, only the change (increase) in relative volume
10. Please explain why the analysis of fat oxidation was performed during drying, when there was still water in the peels in which oxygen was dissolved. Please explain why TBRAS increased during storage? What was the content of unsaturated fats in the peel?
11. What research problem was planned to be solved by examining the changes in the content of individual aroma components during drying? Please explain the reason for the increase in these values ​​during drying
12. The obtained results were not subjected to critical evaluation. The most favorable values ​​of the quality attributes studied were not indicated. The most important evaluation of the obtained crisps, i.e. sensory evaluation, was not performed, without which it is difficult to indicate the most favorable drying parameters
13. Please explain why crispness, after reaching the maximum, decreased despite the decrease in water content.
14. The authors refer to protein oxidation (line 308), but they did not study it?
15. Line 124 drying rate , no drying ratio
Comments on the Quality of English Languagethe language is imprecise
Author Response
|
Response to Reviewer 1 Comments
|
||
|
1. Summary |
|
|
|
Thank you to the reviewing experts for their suggestions on this article. We have carefully read the issues you mentioned and made the necessary modifications, which have been highlighted in red in the attachment. Thank you again for your support and assistance in our work. Your opinions are very important to our research. If you have any other questions or suggestions, please feel free to contact us at any time. |
||
|
2. Questions for General Evaluation |
Reviewer’s Evaluation |
Response and Revisions |
|
Does the introduction provide sufficient background and include all relevant references? |
Can be improved |
|
|
Are all the cited references relevant to the research? |
Must be improved |
|
|
Is the research design appropriate? |
Must be improved |
|
|
Are the methods adequately described? |
Must be improved |
|
|
Are the results clearly presented? |
Must be improved |
|
|
Are the conclusions supported by the results? |
|
|
|
3. Point-by-point response to Comments and Suggestions for Authors |
||
|
Comments 1: Line 124 drying rate , no drying ratio |
||
|
Response 1: Thank you for pointing this out. We agree with this comment. In line 147, we replaced the drying ratio with the drying rate. |
||
|
Comments 2: V3 in formula 8 is not an expansion rate, it is a relative change in volume |
||
|
Response 2:Thank you for pointing this out.We agree with this comment.There is a grammatical error. The ratio should be used to represent the size of the expansion volume, while the rate is used incorrectly.Modifications have been made throughout the entire text |
||
|
Comments 3: Lack of storage conditions described in the methodology |
||
|
Response 3:Thank you for pointing this out. We agree with this comment.Added storage conditions in line 96. |
||
|
Comments 4: Please state in how many repetitions MVD drying was performed. If the results are from only one experiment, without repetitions, then the results are not very reliable |
||
|
Response 4:Thank you for pointing this out. We agree with this comment.This article has conducted more than 3 experiments.The image has been modified. |
||
|
Comments 5: Please state what was the water content in the peel at the beginning and after pre-treatment? |
||
|
Response 5: Thank you for your comment. We have added the data on moisture content, as shown in Figure 2d, in line 299. |
||
|
Comments 6: Drying curves should be described by models based on all experimental data from repetitions and the drying rate should be calculated as a derivative of the function from the best-fit model. The method of calculating the drying rate proposed by the authors is burdened with too large an error |
||
|
Response 6: Thank you for pointing this out. We have carefully checked the calculation method and no mistakes were found, and the similar curves of drying rate were reported in other articles, such as https://doi: 10.1016/j.lwt.2023.115147 |
||
|
Comments 7: Please justify that under microwave heating conditions, water transport is of a diffusion nature (molecular transport). In my opinion, there is a convective movement of steam forced by the pressure difference. Therefore, the law describing the phenomenon of diffusion cannot be applied |
||
|
Response 7: Thank you for pointing this out. For a closed microwave cavity, convection transfer can be ignored and only the diffusion of moisture inside the material can be considered. Similar consideration was reported in other articles, such as, https://doi: 10.1016/j.biosystemseng.2020.05.002 |
||
|
Comments 8: When using a microwave field, there is no absorption of microwaves, there is the setting of water molecules in motion under the influence of a variable electromagnetic field. As a result of friction with neighboring molecules, heat is generated |
||
|
Response 8: Thank you for your valuable comments. We totally agree with your comments. In line 266, the reason for the phenomenon of drying rate is changed to "Friction with adjacent molecules generates heat. |
||
|
Comments 9: The authors do not determine the rate of expansion, only the change (increase) in relative volume |
||
|
Response 9: Thank you for pointing this out. There is a grammatical error. The ratio should be used to represent the size of the expansion volume, while the rate is used incorrectly. Modifications have been made throughout the full-text. |
||
|
Comments 10: Please explain why the analysis of fat oxidation was performed during drying, when there was still water in the peels in which oxygen was dissolved. Please explain why TBRAS increased during storage? What was the content of unsaturated fats in the peel? |
||
|
Response 10: Thank you for pointing this out. Firstly, heating can cause the oxidation of fat, so we measure the degree of fat oxidation at different powers during the drying process. During storage, lipids undergo self oxidation, and the degree of PSC fat oxidation is measured to prepare for extending the product storage period. |
||
|
Comments 11: What research problem was planned to be solved by examining the changes in the content of individual saroma components during drying? Please explain the reason for the increase in these values during drying |
||
|
Response 11: Thank you for pointing this out. The drying process can cause changes in volatile flavor compounds. In the future, we want to explore the relationship between fat oxidation and flavor compound changes. Some literature has shown a certain relationship between aldehyde ketone content and lipid oxidation. For example, this article:http//:doi:10.1016/j.foodres.2024.114772 |
||
|
Comments 12: The obtained results were not subjected to critical evaluation. The most favorable values of the quality attributes studied were not indicated. The most important evaluation of the obtained crisps, i.e. sensory evaluation, was not performed, without which it is difficult to indicate the most favorable drying parameters |
||
|
Response 12: Thank you for pointing this out. We agree with this comment. We supplemented sensory experiments in section 3.2.6. |
||
|
Comments 13: Please explain why crispness, after reaching the maximum, decreased despite the decrease in water content. |
||
|
Response 13: Thank you for pointing this out. Long term drying can significantly damage the internal microstructure of PSC and lead to the collapse of its internal pores, resulting in a decrease in overall brittleness. For example, https://doi: 10.1016/j.lwt.2023.115147 |
||
|
Comments 14: The authors refer to protein oxidation (line 308), but they did not study it? |
||
|
Response 14: Thank you for pointing this out. We totally agree with this comment. Our future work goals regarding protein oxidation issues are not explained in this article. |
||

Reviewer 2 Report
Comments and Suggestions for Authors
The work proposes the study of the drying of pork skins snacks by means of the application of vacuum and microwaves in order to improve the production process of this type of snacks.
The work is interesting, but there are some things to improve that are detailed below:
In l15 reference is made to the Midilli model, it is not necessary to reference it with the et at the end, it is necessary to revise that way of mentioning the model throughout the article.
In l33 reference is made to leisure foods, wouldn't it be better to talk about snacks?
In l38 avoid the use of etc.
In l41 the term expansion is misused or unclear. One could mention the disadvantages in terms of processing time/energy cost/deterioration due to over-processing.
When talking about the use of microwaves, it would be better to talk about microwave and vacuum assisted drying, it is recommended to review the literature on this subject.
In l51 when talking about material expansion, it is necessary to mention that this is due, among other phenomena, to the sudden evaporation that is generated; it is necessary to analyze more rigorously the phenomena associated with this expansion.
In l53 it should be bacterial growth rather than bacterial pollution, in addition it could be mentioned about the mechanisms of heating by irradiation and the associated volumetric heating phenomena. In addition, different final moistures are generally lower, as well as better retention of bioactive compounds due to lower processing times.
In l57-58 we talk about destruction of structures and it should be rather modification.
The wording l84-l87 needs to be revised.
When talking about the objectives, it would be enough to leave only one general objective, as they are, these 3 objectives seem to be taken out of a degree thesis. Revise...
In 2.1 Is there any condition of the raw material such as thickness, and any other characteristic that is important to reference?
l98 revise the wording, it is very colloquial and seems almost like a cooking recipe.
in l101 what do you mean by thirteen spices, it is necessary to specify
In 2.2 it is necessary to specify the charge density, and/or arrangement of the samples, was the device static or did it have movement inside?
Table 1 review Midilli et al
In 2.4.1 Is it necessary to specify the illuminant and the observer used for color measurements?
Revise wording of 2.4.3
2.4.4 Is electronic nose... or volatiles measurement or some other generic concept for the measurement but do not use the name of the apparatus .....
In 2.4.5 the characteristics of the spectrophotometer used are not specified, it is necessary to properly identify all the equipment used.
in l189 it speaks of three copies, does it refer to three replicates?
Review the discussion on the water activity, this should come from the observations and results...it is necessary that the authors review the concepts well and then write their discussion.
It is necessary to revise figure 1 in A, the figure is too small and the numbers are not well seen.
Is it really necessary that the figure is like this with background/depth in b, why do you make the linear adjustment if you then mention that you use the usual models...? Are the 3 figures a b and c really necessary?
Fig 1 c is not discussed, moreover it is necessary to improve the discussion, it is only limited to present the data without making a deeper and more specific analysis in the context of food drying.
l257 repeats information
Table 5 Midilli et al, moreover there is no analysis and meaning of the constants of the models in the context of food drying, improve the discussion of the table.
in l269 it is not clear when talking about combined with fig1.
In the color analysis, qualitative color descriptions are used, if they measured color, which is an objective measurement, stay on that line and do not use color descriptions.
Fig 2 3d plots for what? besides you could have used the delta E calculation, the use of markers with shadows and volumes messes up the plot.
If there are replicates and the values presented are averages, the error bars on all measurements could be added to all plots.
Evaluate if it is possible to improve the presentation of Figure 3.
3.2.5 review the use of the term electronic nose to refer to analysis.
Conclusions should be completely rewritten with a focus on the knowledge generated, its industrial applications and future benefits .....?
Author Response
|
Response to Reviewer 2 Comments |
||
|
1. Summary |
|
|
|
Thank you to the reviewing experts for their suggestions on this article. We have carefully read the issues you mentioned and made the necessary modifications, which have been highlighted in red in the attachment. Thank you again for your support and assistance in our work. Your opinions are very important to our research. If you have any other questions or suggestions, please feel free to contact us at any time. |
||
|
2. Questions for General Evaluation |
Reviewer’s Evaluation |
Response and Revisions |
|
Does the introduction provide sufficient background and include all relevant references? |
Can be improved |
|
|
Are all the cited references relevant to the research? |
Can be improved |
|
|
Is the research design appropriate? |
Can be improved |
|
|
Are the methods adequately described? |
Can be improved |
|
|
Are the results clearly presented? |
Can be improved |
|
|
Are the conclusions supported by the results? |
Must be improved |
|
|
3. Point-by-point response to Comments and Suggestions for Authors |
||
|
Comments 1: In l15 reference is made to the Midilli model, it is not necessary to reference it with the et at the end, it is necessary to revise that way of mentioning the model throughout the article. |
||
|
Response 1: Thank you for pointing this out. The Midilli et al. model was first proposed by Midilli, who then collaborated with their team to simulate and predict the model. Therefore, the ‘’et al’’ was added to this literature, and there are many examples of this in the literature, such as, https://doi: 10.1016/j.lwt.2023.115147, https://doi: 10.1016/j.biosystemseng.2020.05.002 |
||
|
Comments 2: In l33 reference is made to leisure foods, wouldn't it be better to talk about snacks. |
||
|
Response 2: Thank you for pointing this out. We agree with this comment. In the entire text, the leisure foods were replaced by snacks. |
||
|
Comments 3: In l38 avoid the use of etc. Response 3: Thank you for pointing this out. We agree with this comment. In line 36, we delete ect. |
||
|
Comments 4: In l41 the term expansion is misused or unclear. One could mention the disadvantages in terms of processing time/energy cost/deterioration due to over-processing. |
||
|
Response 4: Thank you for pointing this out. We agree with this comment. In lines 40-44, puffing is used instead of expansion, and the disadvantages of frying puffing in terms of processing time/energy cost/poor quality caused by excessive processing are discussed. |
||
|
Comments 5: When talking about the use of microwaves, it would be better to talk about microwave and vacuum assisted drying, it is recommended to review the literature on this subject. |
||
|
Response 5: Thank you for pointing this out. We agree with this comment. We first discussed microwave drying, and then explained the advantages of microwave drying and vacuum assisted drying in lines 56-59. |
||
|
Comments 6: In l51 when talking about material expansion, it is necessary to mention that this is due, among other phenomena, to the sudden evaporation that is generated; it is necessary to analyze more rigorously the phenomena associated with this expansion. |
||
|
Response 6: Thank you for pointing this out. We agree with this comment. Explained the phenomenon of material puffing in line 52-56. |
||
|
Comments 7: In l53 it should be bacterial growth rather than bacterial pollution, in addition it could be mentioned about the mechanisms of heating by irradiation and the associated volumetric heating phenomena. In addition, different final moistures are generally lower, as well as better retention of bioactive compounds due to lower processing times. |
||
|
Response 7: Thank you for pointing this out. We agree with this comment. In line 60, we replaced bacterial pollution with bacterial growth. Described the better retention of PSC bioactive compounds. |
||
|
Comments 8: In l57-58 we talk about destruction of structures and it should be rather modification |
||
|
Response 8: Thank you for pointing this out. We agree with this comment. In line 63, we replaced destruction by modification. |
||
|
Comments 9: The wording l84-l87 needs to be revised. |
||
|
Response 9: Thank you for pointing this out. We agree with this comment. We have revised the paragraph at line 91-95. |
||
|
Comments 10: When talking about the objectives, it would be enough to leave only one general objective, as they are, these 3 objectives seem to be taken out of a degree thesis. Revise... |
||
|
Response 10: Thank you for pointing this out. We agree with this comment. In line 95, the objectives were revised. |
||
|
Comments 11: In 2.1 Is there any condition of the raw material such as thickness, and any other characteristic that is important to reference? |
||
|
Response 11: Thank you for pointing this out. We agree with this comment. The thickness of the pigskin was added in line 106, and the location of the pigskin was explained in line 100. |
||
|
Comments 12: l98 revise the wording, it is very colloquial and seems almost like a cooking recipe |
||
|
Response 12: Thank you for pointing this out. We agree with this comment. This formula is associated with the subsequent changes in flavor and has been added in line 107 as' The specific formula for steaming pig skin is as follows'. |
||
|
Comments 13: in l101 what do you mean by thirteen spices, it is necessary to specify |
||
|
Response 13: Thank you for pointing this out. Replace with various seasonings in the article, this is a writing error, sorry. |
||
|
Comments 14: In 2.2 it is necessary to specify the charge density, and/or arrangement of the samples, was the device static or did it have movement inside? |
||
|
Response 14: Thank you for pointing this out. We agree with this comment. The arrangement of the samples is described in the form of images in line 113, as shown in Figure 1, with the device in a stationary state. Supplementary information is provided in line 113. |
||
|
Comments 15: Table 1 review Midilli et al |
||
|
Response 15: Thank you for pointing this out. The Midilli et al. model was first proposed by Midilli, who then collaborated with their team to simulate and predict the model. Therefore, the “et al.” was added to this literature, and there are many examples of this in the literature, such as https://doi: 10.1016/j.lwt.2023.115147, https://doi: 10.1016/j.biosystemseng.2020.05.002 |
||
|
Comments 16: In 2.4.1 Is it necessary to specify the illuminant and the observer used for color measurements? |
||
|
Response 15: The color difference machine is a specific machine, as shown in the following figure.
|
||
|
Comments 16: Revise wording of 2.4.3 |
||
|
Response 16: Thank you for pointing this out. We agree with this comment. Thank you for pointing this out. We agree with this comment. There is a grammatical error. The ratio should be used to represent the size of the expansion volume, while the rate is used incorrectly. The modifications have been made throughout the entire text. |
||
|
Comments 17: 2.4.4 Is electronic nose... or volatiles measurement or some other generic concept for the measurement but do not use the name of the apparatus ..... |
||
|
Response 17: Thank you for pointing this out. We agree with this comment. The general concept of electronic nose was explained in 237-238. Quoting the writing style of the article. https://doi: 10.1016/j.foodres.2019.05.041 |
||
|
Comments 18: In 2.4.5 the characteristics of the spectrophotometer used are not specified, it is necessary to properly identify all the equipment used. |
||
|
Response 18: Thank you for pointing this out. We agree with this comment. The mode of instrument was re-added in line 209. |
||
|
Comments 19: In l189 it speaks of three copies, does it refer to three replicates? |
||
|
Response 19: Thank you for pointing this out. We agree with this comment. In line 227,we replaced copies with replicates. |
||
|
Comments 20: Review the discussion on the water activity, this should come from the observations and results...it is necessary that the authors review the concepts well and then write their discussion. |
||
|
Response 20: Thank you for pointing this out. We agree with this comment. In 237-238, two drying stages were explained. |
||
|
Comments 21: It is necessary to revise figure 1 in A, the figure is too small and the numbers are not well seen. |
||
|
Response 21: Thank you for pointing this out. The images have been modified as shown in Figure 2a. |
||
|
Comments 22: Is it really necessary that the figure is like this with background/depth in b, why do you make the linear adjustment if you then mention that you use the usual models...? Are the 3 figures a b and c really necessary? |
||
|
Response 22: Thank you for pointing this out. We agree with this comment. The images have been modified as shown in Figure 2b. The linear adjustment here is from formula (5), which calculates the Deff of PSC. |
||
|
Comments 23: Fig 1 c is not discussed, moreover it is necessary to improve the discussion, it is only limited to present the data without making a deeper and more specific analysis in the context of food drying. |
||
|
Response 23: Thank you for pointing this out. We agree with this comment. According to the linear fitting in the figure, Table 5 is obtained. |
||
|
Comments 24: l257 repeats information |
||
|
Response 24: Thank you for pointing this out. We agree with this comment. The sentence 'This also confirms the above viewpoint' was added in line 315. |
||
|
Comments 25: Table 5 Midilli et al, moreover there is no analysis and meaning of the constants of the models in the context of food drying, improve the discussion of the table. |
||
|
Response 25: Thank you for pointing this out. We agree with this comment. These constants are numerical values obtained through model fitting, and the model fitting effect is mainly analyzed through R2, chi square, and RMSE. Currently, no literature has been found to analyze the constants. |
||
|
Comments 26: Table 5 Midilli et al, moreover there is no analysis and meaning of the constants of the models in the context of food drying, improve the discussion of the table. |
||
|
Response 26: Thank you for pointing this out. We agree with this comment. At line 336, the description of Fig.1 was removed and a new description was provided. |
||
|
Comments 27: If there are replicates and the values presented are averages, the error bars on all measurements could be added to all plots. |
||
|
Response 27: Thank you for pointing this out. We agree with this comment. The image has been modified as shown in Figure 3. |
||
|
Comments 28: Evaluate if it is possible to improve the presentation of Figure 3. |
||
|
Response 28: Thank you for pointing this out. We agree with this comment. Representing the graph in a table, it was found that the table was too large, so it was used to represent it. |
||
|
Comments 29:3.2.5 review the use of the term electronic nose to refer to analysis. |
||
|
Response 29: Thank you for pointing this out. In most articles, the expression of electronic nose is used. https://doi: 10.1016/j.foodres.2019.05.041 |
||
|
Comments 30: Conclusions should be completely rewritten with a focus on the knowledge generated, its industrial applications and future benefits .....? |
||
|
Response 30: Thank you for pointing this out. We agree with this comment. The conclusion has been revised in conjunction with industrial applications and future benefits |
||

Reviewer 3 Report
Comments and Suggestions for Authors
Please, see the attached file.

Author Response
|
Response to Reviewer 3 Comments
|
||
|
1. Summary |
|
|
|
Thank you to the reviewing experts for their suggestions on this article. We have carefully read the issues you mentioned and made the necessary modifications, which have been highlighted in red in the attachment. Thank you again for your support and assistance in our work. Your opinions are very important to our research. If you have any other questions or suggestions, please feel free to contact us at any time. |
||
|
2. Questions for General Evaluation |
Reviewer’s Evaluation |
Response and Revisions |
|
Does the introduction provide sufficient background and include all relevant references? |
Can be improved |
|
|
Are all the cited references relevant to the research? |
Can be improved |
|
|
Is the research design appropriate? |
Yes |
|
|
Are the methods adequately described? |
Yes |
|
|
Are the results clearly presented? |
Yes |
|
|
Are the conclusions supported by the results? |
Yes |
|
|
3. Point-by-point response to Comments and Suggestions for Authors |
||
|
Comments 1: Just a point to be highlighted, they use for example, ‘at present’ and ‘recently’ when they are talking about papers of 2012, 2016 and 2019. Please, modify these points. |
||
|
Response 1: Thank you for pointing this out. We agree with this comment. ‘at present’ and ‘recently’ have been deleted and the original text has been modified on line 30,34 |
||
|
Comments 2: In the introduction section, lacks the demand of the consumers about this kind of products. Otherwise, why they develop these products? |
||
|
Response 2: Thank you for pointing this out. The article introduces the advantages of pigskin and the attention paid to puffed foods, so we want to make a pigskin puffed food to enrich the food market. |
||
|
Comments 3: L* component respect del CIELAB means lightness but not brightness. Please, check this point. |
||
|
Response 3: Thank you for pointing this out. In many literature, L* is used to represent brightness. For example, this article: https://doi: 10.1016/j.lwt.2023.115147. |
||
|
Comments 4: Authors deal with the measure of colour but in section 2.4.1 they don’t specify samples used for that. |
||
|
Response 4: Thank you for pointing this out. the sentence 'Measure the color of PSC at different times under 600 W, 700 W, and 800 power' was added in line 176-177. |
||
|
Comments 5: Section dealing with the instrumental texture analysis specify the use of PSC test but authors don’t explain which is this test and they relationship with the sensory measures. |
||
|
Response 5: Thank you for pointing this out. The sensory evaluation analysis of PSC in the article was added. |
||
|
Comments 6: Please, check the format of the citations in the text in order to follow the recommendations of Foods. |
||
|
Response 6: Thank you for pointing this out. The sensory evaluation analysis of PSC in the article was added. |
||
|
Comments 7: Figures should be improved. It’s very difficult to understand if there are significant differences between treatments. |
||
|
Response 7: Thank you for pointing this out. We have made modifications to the images and optimized their saliency. |
||
|
Comments 8: In general, no relationship among any of the instrumental parameters measured and the sensory attributes of the food product are mentioned through the paper. This point is a constraint for this paper. Food is intended to be consumed, hence the importance of including sensory analysis in the paper. |
||
|
Response 8: Thank you for pointing this out. The sensory evaluation analysis of PSC was added in the article. |
||

Reviewer 4 Report
Comments and Suggestions for Authors
The article “Evaluation of Drying Characteristics and Quality Attributes for 2 Microwave Vacuum Drying of Pork Skin Crisps” has been reviewed and the following comments have been obtained for the improvement of the manuscript.
In the introduction section, it is necessary to specify what quantity or percentage of pork skin is wasted, to have a perspective of the importance of this study. Pork skin is not wasted in other regions of the world. So it is important to put it in context.
It is possible that this document will help to reinforce your introduction and discussion of results: https://doi.org/10.3390/pr12091969
In the methodology section, justify why it is boiled before being dried and then baked. What are the recommended humidity levels for each process.
Was the temperature of the food measured during microwave dehydration? Was any cooking process observed?
Figure 1a, it is necessary to increase the font size to better observe the experimental data. Better discuss the drying speed from a phenomenological point of view.
It would be interesting to add some photos of the products before and after drying, to observe how this change occurred. Try to include a comparison with some other similar works.
In general, improve the quality of the graphics regarding the size of the letter. The conclusions are adequate.
Author Response
|
Response to Reviewer 4 Comments
|
||
|
1. Summary |
|
|
|
Thank you to the reviewing experts for their suggestions on this article. We have carefully read the issues you mentioned and made the necessary modifications, which have been highlighted in red in the attachment. Thank you again for your support and assistance in our work. Your opinions are very important to our research. If you have any other questions or suggestions, please feel free to contact us at any time. |
||
|
2. Questions for General Evaluation |
Reviewer’s Evaluation |
Response and Revisions |
|
Does the introduction provide sufficient background and include all relevant references? |
Yes |
|
|
Are all the cited references relevant to the research? |
Yes |
|
|
Is the research design appropriate? |
Yes |
|
|
Are the methods adequately described? |
Yes |
|
|
Are the results clearly presented? |
Yes |
|
|
Are the conclusions supported by the results? |
Yes |
|
|
3. Point-by-point response to Comments and Suggestions for Authors |
||
|
Comments 1: In the introduction section, it is necessary to specify what quantity or percentage of pork skin is wasted, to have a perspective of the importance of this study. Pork skin is not wasted in other regions of the world. So it is important to put it in context. |
||
|
Response 1: Thank you for pointing this out. We agree with this comment. In line 30, the food plasticity of pig by-products was described, eliminating the notion of waste. |
||
|
Comments 2: It is possible that this document will help to reinforce your introduction and discussion of results: https://doi.org/10.3390/pr12091969 |
||
|
Response 2: Thank you for pointing this out. I have carefully read this article and it has been very helpful to me, and this document was cited in this revised article. |
||
|
Comments 3: In the methodology section, justify why it is boiled before being dried and then baked. What are the recommended humidity levels for each process. |
||
|
Response 3: Thank you for pointing this out. Boiling before drying can further remove the oil from the pigskin, and the heating process makes the structure softer, which is beneficial for microwave experiments. At the same time, seasonings can also be added during the cooking process to enhance the sensory experience of pig skin. The indoor humidity during the experiment was around 65%. |
||
|
Comments 3: Was the temperature of the food measured during microwave dehydration? Was any cooking process observed? |
||
|
Response 3: Thank you for pointing this out. The indoor temperature during the experiment was around 25 ℃. The operation process is in a closed space, and the cooking process of PSC cannot be seen. |
||
|
Comments 4: Figure 1a, it is necessary to increase the font size to better observe the experimental data. Better discuss the drying speed from a phenomenological point of view. |
||
|
Response 4: Thank you for pointing this out. We agree with this comment. The images have been modified as shown in Figure 2a. |
||
|
Comments 5: It would be interesting to add some photos of the products before and after drying, to observe how this change occurred. Try to include a comparison with some other similar works. |
||
|
Response 5: Thank you for pointing this out. We have added a photo of the tray arrangement, as shown in Figure 1. |
||
|
Comments 6: It would be interesting to add some photos of the products before and after drying, to observe how this change occurred. Try to include a comparison with some other similar works. |
||
|
Response 6: Thank you for pointing this out. We agree with this comment. We have optimized the images. |
||

Round 2
Reviewer 1 Report
Comments and Suggestions for Authors
The authors have taken into account most of my comments. However, I do not agree with the way of presenting the drying curve and determining the drying rate. You cannot connect the measurement points with a straight line, because the real changes are not rectilinear. If you do this, you get such nonsense that for 700 W, the drying rate increases fourfold between the fifth and sixth minute of drying, and then decreases to zero in the next minute. The measurement points determine the trend line and it illustrates the course of the process. The fact that some magazine accepted the presentation of the drying curves in the form presented by the authors for printing is no argument for me.
In Figure 2 a and b and c and d, the same color of the line should be maintained for a given microwave power. Changing the color of the line is misleading
Author Response
|
Response to Reviewer 1 Comments
|
||
|
1. Summary |
|
|
|
Thank you to the reviewing experts for their suggestions on this article. We have carefully read the issues you mentioned and made the necessary modifications, which have been highlighted in red in the attachment. Thank you again for your support and assistance in our work. Your opinions are very important to our research. If you have any other questions or suggestions, please feel free to contact us at any time. |
||
|
2. Questions for General Evaluation |
Reviewer’s Evaluation |
Response and Revisions |
|
Does the introduction provide sufficient background and include all relevant references? |
Yes |
|
|
Are all the cited references relevant to the research? |
Yes |
|
|
Is the research design appropriate? |
Yes |
|
|
Are the methods adequately described? |
Must be improved |
|
|
Are the results clearly presented? |
Yes |
|
|
Are the conclusions supported by the results? |
Yes |
|
|
3. Point-by-point response to Comments and Suggestions for Authors |
||
|
Comments 1: The authors have taken into account most of my comments. However, I do not agree with the way of presenting the drying curve and determining the drying rate. You cannot connect the measurement points with a straight line, because the real changes are not rectilinear. If you do this, you get such nonsense that for 700 W, the drying rate increases fourfold between the fifth and sixth minute of drying, and then decreases to zero in the next minute. The measurement points determine the trend line and it illustrates the course of the process. The fact that some magazine accepted the presentation of the drying curves in the form presented by the authors for printing is no argument for me. |
||
|
Response 1: Thank you for pointing this out. We agree with this comment. We have made changes to the image by removing the line effect. |
||
|
Comments 2: In Figure 2 a and b and c and d, the same color of the line should be maintained for a given microwave power. Changing the color of the line is misleading |
||
|
Response 2: Thank you for pointing this out. We agree with this comment. The color of the straight lines in the image has been unified. |
||

Reviewer 2 Report
Comments and Suggestions for Authors
The efforts of the authors are gratefully acknowledged. They have provided satisfactory answers to most of my observations.
Author Response
|
Response to Reviewer 2 Comments
|
||
|
1. Summary |
|
|
|
Thank you to the reviewing experts for their suggestions on this article. We have carefully read the issues you mentioned and made the necessary modifications, which have been highlighted in red in the attachment. Thank you again for your support and assistance in our work. Your opinions are very important to our research. If you have any other questions or suggestions, please feel free to contact us at any time. |
||
|
2. Questions for General Evaluation |
Reviewer’s Evaluation |
Response and Revisions |
|
Does the introduction provide sufficient background and include all relevant references? |
Yes |
|
|
Are all the cited references relevant to the research? |
Yes |
|
|
Is the research design appropriate? |
Yes |
|
|
Are the methods adequately described? |
Yes |
|
|
Are the results clearly presented? |
Yes |
|
|
Are the conclusions supported by the results? |
Yes |
|
|
3. Point-by-point response to Comments and Suggestions for Authors |
||
|
Comments 1: The efforts of the authors are gratefully acknowledged. They have provided satisfactory answers to most of my observations. |
||
|
Response 1: Thank you for the reviewer's efforts. We have revised the article according to your requirements, and your suggestions have made our article more complete and fluent. |
||
Reviewer 3 Report
Comments and Suggestions for Authors
the authors may replace 'taste' by flavour
Author Response
|
Response to Reviewer 3 Comments
|
||
|
1. Summary |
|
|
|
Thank you to the reviewing experts for their suggestions on this article. We have carefully read the issues you mentioned and made the necessary modifications, which have been highlighted in red in the attachment. Thank you again for your support and assistance in our work. Your opinions are very important to our research. If you have any other questions or suggestions, please feel free to contact us at any time. |
||
|
2. Questions for General Evaluation |
Reviewer’s Evaluation |
Response and Revisions |
|
Does the introduction provide sufficient background and include all relevant references? |
Yes |
|
|
Are all the cited references relevant to the research? |
Yes |
|
|
Is the research design appropriate? |
Yes |
|
|
Are the methods adequately described? |
Yes |
|
|
Are the results clearly presented? |
Yes |
|
|
Are the conclusions supported by the results? |
Yes |
|
|
3. Point-by-point response to Comments and Suggestions for Authors |
||
|
Comments 1: The authors may replace 'taste' by flavour |
||
|
Response 1: Thank you for pointing this out. We agree with this comment. In the entire text, we use “flavor” instead of “taste”. |
||
